# Efficient Synthesis of 2-OH Thioglycosides from Glycals Based on the Reduction of Aryl Disulfides by NaBH_4_

**DOI:** 10.3390/molecules27185980

**Published:** 2022-09-14

**Authors:** Yang-Fan Guo, Tao Luo, Guang-Jing Feng, Chun-Yang Liu, Hai Dong

**Affiliations:** Key Laboratory of Material Chemistry for Energy Conversion and Storage, Ministry of Education, Hubei Key Laboratory of Material Chemistry and Service Failure, School of Chemistry & Chemical Engineering, Huazhong University of Science & Technology, Luoyu Road 1037, Wuhan 430074, China

**Keywords:** glycals, 2-OH thioglycosides, glycosyl donors, glycosylation, one-pot reaction

## Abstract

An improved method to efficiently synthesize 2-OH thioaryl glycosides starting from corresponding per-protected glycals was developed, where 1,2-anhydro sugars were prepared by the oxidation of glycals with oxone, followed by reaction of crude crystalline 1,2-anhydro sugars with NaBH_4_ and aryl disulfides. This method has been further used in a one-pot reaction to synthesize glycosyl donors having both “armed” and “NGP (neighboring group participation)” effects.

## 1. Introduction

The chemical synthesis of complex carbohydrates is an important research topic in carbohydrate chemistry due to their crucial roles in biological processes [1,2,3,4,5]. Thioglycoside donors are widely used in these syntheses due to their advantages, such as easy preparation, stable chemical properties, and various activation methods [6,7,8,9,10,11,12]. Generally, each glycosylation reaction between the donor and acceptor may produce a product in the α or β configuration, which will lead to adverse effects, such as reduced yield and difficult purification. Therefore, many studies have been devoted to solving the problem of stereoselectivity in glycosylation. Among them, the use of neighboring group participation (NGP) from 2-positions of glycoside donors to control stereoselectivity is a very effective method. The acyl group at 2-position is the most commonly used “NGP” group, which leads to the 1,2-*trans*-configuration of the products in glycosylation [13,14,15,16,17]. However, the “disarmed” effect of acyl groups often leads to low reactivity of glycosyl donors in glycosylation. Therefore, several methods using NGP from 2-ether groups of thioglycoside donors to control stereoselectivity have been developed recently, and phenyl-3,4,6-tri-*O*-benzyl-1-thio-β-D-glucopyranoside **1** is often the precursor required for the synthesis of these donors (Figure 1) [18,19,20,21,22].

Although excellent stereoselectivity and high reactivity were shown in these methods, the synthesis of **1** is a challenge, which reduces the practicality of these methods. For example, the traditional “orthoester method” requires multiple-step protection and deprotection, leading to low synthesis efficiency (Figure 2a) [23,24,25]. More efficient methods involved the formation of 1,2-anhydro sugars by oxidation of glycals with oxone in acetone and the installation of the 1-thiophenyl group through ring-opening reactions of these 1,2-anhydro sugars. For example, the oxidation of glucal **2** by oxone yielded 1,2-anhydro glucose **3** (crude crystalline product): (a) the ring opening of **3** led to 43% (the use of TBASPh) [26] or 47% (the use of NaSPh) [27] yield of **1** in the presence of phenylthiolate at room temperature for overnight, and (b) the ring opening of **3** led to 37–55% yield [19,20] of **1** in the presence of PhSH and ZnCl_2_ at room temperature for overnight (Figure 2b). In this study, an improved method for the efficient synthesis of 2-OH, 1-thioaryl glycosides was developed, in which the ring opening reaction of 1,2-anhydro sugars occurred in the presence of NaBH_4_ and alkyl disulfides at room temperature. As a result, 73% yield of **1** could be efficiently prepared from **2** (two-step reaction was completed within 90 min) under very mild conditions (Figure 2c). Furthermore, the glycosyl donors having both “armed” and “NGP” effects can be efficiently synthesized in a one-pot reaction based on this method.

## 2. Results

In our lab, we were working on developing relative green methods for 1-thioglycosides by avoiding the use of odorous thioacetic acids and alkylthiols [28,29,30]. We noticed a report where NaBH_4_ and disulfides were used instead of sodium arylthiolates in the synthesis of 1-thioglycosides [31]. It was observed that phenylselenolate and phenylthiolate were quickly generated by mixing diselenide or disulfide precursors with a stoichiometric amount of NaBH_4_ in acetonitrile (Reaction formula shown in Figure 3a,b). This inspired us to explore whether a system of disulfides and NaBH_4_ could be used to improve the synthesis of **1** [26,27] starting from glucal **2**.

Thus, 1,2-anhydro glucose **3** was first prepared by oxidation of glycal **2** with oxone in acetone, and then, its crude crystals were directly reacted with 0.7 equiv of phenyl disulfide and 1.5 equiv of NaBH_4_ (equivalent to 1.4 equiv of NaBH_3_SPh) at rt in acetonitrile for 1 h to yield **1** in 72% yield, yielding **1** in 75% yield when the reaction was performed at 0 °C for 4 h (entry 1 in Table 1). Due to concerns about direct hydrogenation reduction of NaBH_4_ to **3**, we first allowed NaBH_4_ to react with phenyl disulfide at 50 °C in acetonitrile for 1 h and then added crude crystalline **3** to the reaction mixture (entry 2). Yielding **1** in 73% yield indicated that we had been overly concerned about the possible side effects caused by NaBH_4_. Reducing the amount of NaBH_4_ to 1.0 equiv (equivalent to 1.0 equiv of NaBH_3_SPh and 0.4 equiv of HSPh) resulted in a decrease in the yield of **1** to 68%, and reducing the amount of NaBH_4_ to 0.7 equiv (equivalent to 0.7 equiv of NaBH_3_SPh and 0.7 equiv of HSPh) resulted in a decrease in the yield of **1** to 60% (entry 3). The use of 1.0/2.0 equiv of NaBH_4_ and 0.5/1.0 equiv of phenyl disulfide (equivalent to 1.0/2.0 equiv of NaBH_3_SPh) led to 55%/65% yield of **1** (entry 4). The use of 1.2 equiv of NaBH_4_ and 0.6/0.8 equiv of phenyl disulfide led to 65%/69% yield of **1** (entry 5). These results suggested that 1.5 equiv of NaBH_4_ and 0.7 equiv of phenyl disulfide should be the optimal conditions. We also examined the effect of solvents (acetone, DMF, MeOH, and DCM) on the reaction (entries 6 and 7). These results suggested that acetonitrile should be the optimal solvent. As a comparison, we allowed **3** to react with 1.5 equiv of NaSPh at rt in acetonitrile, which gave **1** in 43% yield after 36 h, indicating the low reactivity of this reaction (entry 8).

We proposed the mechanism of the reaction between **3** and NaBH_3_SPh in Figure 3c and the mechanism of the reaction between **3** and NaSPh in Figure 3d. The coordination of the boron atom of borane instead of Na^+^ (or H^+^) to the 1,2-anhydro oxygen atom may greatly enhance the nucleophilic attack activity of ^−^SPh towards the 1-position of **3**, which explains the results shown in entries 1, 2, and 8 in Table 1. The results shown in entry 3 in Table 1 suggested that NaBH_3_SPh may catalyze the reaction of **3** with HSPh. A possible catalytic mechanism is shown in Figure 3c,e where NaBH_3_SPh reacts with **3** to form **1a** (Figure 3c) and regenerates (Figure 3e) from the exchange of [NaBH_3_]^+^ and H^+^ (Na^+^) between **1a** and HSPh (or NaSPh). Compared to the result (**1**, 42%, 36 h) shown in entry 8, the result (**1**, 55%, 4 h) shown in entry 9 showed that the use of 0.1 equiv of NaBH_4_ and 0.1 equiv of phenyl disulfide (equivalent to 0.1 equiv of NaBH_3_SPh and 0.1 equiv of HSPh) in the presence of 1.2 equiv of NaSPh (1.4 equiv of ^−^SPh existing in the system) led to higher reactivity. The optimal conditions were the use of 0.3 equiv of NaBH_4_, 0.3 equiv of phenyl disulfide and 0.8 equiv of NaSPh (1.4 equiv of ^−^SPh existing in the system also), by which 72% yield of **1** was obtained after 4 h’ reaction (entry 10); continuing to increase the amount of NaBH_4_ and phenyl disulfide to 0.5 equiv (using 0.4 equiv of NaSPh in order to keep 1.4 equiv of ^−^SPh present in the system) instead reduced the yield of **1** to 65% (entry 11). Since aryl disulfides are generally more commercially available reagents than sodium arylthiolates, the conditions shown in entries 1–2 are obviously more practical than that shown in entry 10.

With the optimized conditions in hand, we next set out to evaluate this method using phenyl disulfide with various glycals as substrates (Figure 4). As can be seen, phenyl-2-OH-1-thio-β-D-glucopyranosides **4**–**11** and phenyl-2-OH-1-thio-β-D-galactopyranosides **12**–**14** were efficiently synthesized in 50–70% yields starting from the corresponding glucals and galactals with various protecting groups. For compounds **4**, **5**, **6**, **13**, and **14**, the TBS, acetyl, or benzoyl can be removed orthogonally in the presence of benzyl-protecting group under corresponding acid–base conditions. Thus, these compounds can be used as building blocks for the elongation of sugar chains and the synthesis of branched oligosaccharides. Phenyl-2-OH-3,4-di-OBn-1-thio-β-D-xylopyranoside **15** was synthesized in 65% yield from 3,4-di-OBn xylal, and phenyl-2-OH-1-thio-β-D-lactoside **16** was synthesized in 56% yield from per-benzylated lactal. These results suggested that this method should be applicable to various glycals.

We next evaluated this method using various disulfides with glucal **2** as the substrate (Figure 5). As can be seen, aryl disulfides worked well in this method, leading to 2-OH, β-D-thioglucosides **17**–**22** in 70–76% yields, but non-aryl disulfides did not. The 2-OH, β-D-thioglucosides **23**–**25** could not be obtained by this method. Hydroreduction product **27** was isolated in 26–37% yield in the reaction with non-aryl disulfides, indicating that NaBH_4_ had not been consumed by the reaction with non-aryl disulfides. Further experiments indicated that NaBH_4_ could not reduce non-aryl disulfides even at 50 °C. 

Phenyl diselenide also worked well in this method, leading to 2-OH, β-D-selenoglucoside **26** in 61% yield. However, this reaction took a long time due to the low reactivity for reduction of diselenide by NaBH_4_. In light of the mechanism shown in Figure 3c, we speculated that **1a** should be able to react directly with RX (X represents ^−^Cl or ^−^Br) in the present of NaH to form various thioglycoside donors containing “NGP” group at their 2-positions. This speculation was supported by further experiments and a one-pot method was developed by us (Figure 6). Once the TLC plate showed complete consumption of 1,2-anhydro sugar, NaH and RX were added to the reaction mixture, and the reaction proceeded at rt for 1–4 h, leading to thioglycoside donors **28**–**34** in 48–68% yields based on glycals, respectively. It has been reported that 2-Pic STaz-donors exhibited good reactivity and steroselectivity in glycosylation with Cu(OTf)_2_ as promoter (2-Pic glucoside STaz-donor was obtained in 60% yield over four steps from orthoester) [10], while 2-Pic glucoside SEt-donor exhibited no reactivity with NIS/TfOH as promoter [10b]. We then evaluated the glycosylation between 2-Pic SPh-donors **28**/**29** and various acceptors with NIS/TfOH as promoter (Figure 7). As can be seen, disaccharides **35**–**41** with absolute β-configuration were obtained in 50–86% yields.

## 3. Conclusions

In conclusion, in order to efficiently obtain thioglycoside donors whose protecting groups at 2-position have both “armed” and “NGP” effects, we developed an efficient method for the synthesis of 2-OH thioaryl glycosides starting from their corresponding glycals. In this method, the oxidation of glycals with oxone led to 1,2-anhydro sugars, which are easily isolated by crystallization, and the obtained crude crystalline 1,2-anhydro sugars were then treated with 1.5 equiv of NaBH_4_ and 0.7 equiv of aryl disulfides in acetonitrile at mild conditions to yield the corresponding 1-thioaryl glycosides with 2-OH in 50–75% total yields. Based on this method, thioglycoside donors having both “armed” and “NGP” effects can be efficiently synthesized in a one-pot reaction. Compared with previous methods [19,20,26,27], this method shows three outstanding advantages: good yields, high synthesis efficiency, and the use of relatively green reagents (avoiding the use of foul-smelling aryl thiol reagents).

## 4. Materials and Methods

**General Methods.** All commercially available starting materials and solvents were of reagent grade and used without further purification. Chemical reactions were monitored with thin-layer chromatography using precoated silica gel 60 (0.25 mm thickness) plates. Flash column chromatography was performed on silica gel 60 (SDS 0.040–0.063 mm). ^1^H NMR spectra were recorded at 298 K in CDCl_3_ using the residual signals from CHCl_3_ (^1^H: = 7.26 ppm) as internal standard. ^1^H peak assignments were made by first order analysis of the spectra, supported by standard ^1^H-^1^H correlation spectroscopy (COSY) (see Appendix A).

**General process A for synthesis of 2-OH 1-thioaryl glycosides from glycals.***Step 1*. To a cooled (0 °C) solution of a per-protected glycal (1 mmol) in DCM (4 mL) were added acetone (0.4 mL) and saturated aqueous NaHCO_3_ (7 mL). The mixture was stirred vigorously, and a solution of oxone (2 mmol) in H_2_O (2.5 mL) was added dropwise over 10 min. The mixture was stirred vigorously at 0 °C for 30 min and then at rt until TLC indicated consumption of the starting material. The organic phase was separated, and the aqueous phase was extracted with DCM (2 × 10 mL). The combined organic phases were dried (MgSO_4_) and concentrated in vacuo to obtain the crude 1,2-anhydro sugar.

*Step 2*. To a mixture of phenyl disulfide (or phenyl diselenide) (0.7 mmol) and NaBH_4_ (53 mg, 1.4 mmol) was added acetonitrile (5 mL). The mixture was stirred at rt for 30 min to 2 h until TLC indicated full conversion of the phenyl disulfide (or phenyl diselenide). The mixture was then added to the crude α-1,2-anhydro sugars. The reaction was stirred at rt for 5–60 min until TLC indicated full conversion of the starting material. The mixture was diluted with DCM and washed with water. The aqueous phase was re-extracted with DCM, and collected organic phases were dried and evaporated under vacuum. The residue was purified by silica gel flash chromatography.


**General process B for one-pot synthesis of thioglycoside donors containing a “NGP” group at the 2-position.**


*Step 1*. Same as *step 1* in general process A.

*Step 2*. To a mixture of phenyl disulfide (145 mg, 0.7 mmol) and NaBH_4_ (53 mg, 1.4 mmol) was added acetonitrile (5 mL). The mixture was stirred at rt for 30 min to 2 h until TLC indicated full conversion of the phenyl disulfide. The mixture was then added to the crude α-1,2-anhydro sugars. The reaction was stirred at rt for 5–60 min until TLC indicated full conversion of the starting material. The reaction mixture was then cooled to 0 °C, followed the slow addition of sodium hydride (6.0 mmol, 6 equiv, 60% oil dispersion), and allowed to stir at 0 °C for 10 min. After that, alkylation/acylation reagents (2–3 equiv) were added to the reaction mixture. The reaction mixture was allowed to warm to rt and then stirred for 1–4 h. Upon completion, the reaction was quenched by adding crushed ice (10 g), stirred until cessation of H_2_ evolution, and then extracted with ethyl acetate (3 × 80 mL). The combined organic phase was washed with water (3 × 40 mL), separated, dried with MgSO_4_, and evaporated in vacuo. The residue was purified by column chromatography.

**General process C for typical NIS/TfOH-promoted glycosylation procedure.** A mixture of a glycosyl donor (0.13 mmol), a glycosyl acceptor (0.10 mmol), and freshly activated molecular sieves (4 Å, 200 mg) in CH_2_Cl_2_ (1.6 mL) was stirred under an atmosphere of argon for 1 h. After NIS (0.26 mmol) and TfOH (0.013 mmol) were added at −25 °C, the reaction mixture was allowed to warm to rt over 1 h and then was quenched with TEA and stirred for 30 min. The mixture was then diluted with CH_2_Cl_2_, the solid was filtered-off, and the residue was washed with CH_2_Cl_2_. After the combined filtrate (30 mL) was washed with water (4 × 10 mL), the organic phase was separated, dried with MgSO_4_,and concentrated in vacuo. The residue was purified by silica gel flash chromatography.

*Phenyl 3*,*4*,*6-tri-O-benzyl-1-thio-β-D-glucopyranoside (**1**)* [20]. Following general process A, starting from **2** (100 mg, 0.24 mmol), after 1 h of ring-opening reaction for the crude anhydro sugar, purification by silica gel flash column chromatography afforded **1** as a white solid (94 mg, 72%). Rf = 0.43 (petroleum ether/ethyl acetate 4:1); ^1^H NMR (600 MHz, chloroform-*d*) δ 7.54–7.64 (m, 2H), 7.42–7.25 (m, 16H), 7.24–7.19 (m, 2H), 4.94 (d, *J* = 11.2 Hz, 1H), 4.90–4.81 (m, 2H), 4.67–4.55 (m, 3H), 4.53 (d, *J* = 9.6 Hz, 1H), 3.83 (dd, *J* = 11.0, 2.0 Hz, 1H), 3.77 (dd, *J* = 11.0, 4.5 Hz, 1H), 3.67–3.59 (m, 2H), 3.58–3.44 (m, 2H), 2.43 (s, 1H) ppm.

*Phenyl 3*,*4-di-O-benzyl-6-O-tert-butyl-dimethylsily-1-thio-β-D-glucopyranoside (**4**).* Following general process A, starting from **4a** (50 mg, 0.114 mmol), after 1 h of ring-opening reaction for the crude anhydro sugar, purification by silica gel flash column chromatography afforded **4** as a colorless oil (40 mg, 64%). Rf = 0.41 (petroleum ether/ethyl acetate 8:1); ^1^H NMR (400 MHz, chloroform-*d*) δ 7.51–7.43 (m, 2H), 7.31–7.17 (m, 13H), 4.85–4.73 (m, 3H), 4.60 (d, *J* = 10.8 Hz, 1H), 4.39 (d, *J* = 9.6 Hz, 1H), 3.86–3.74 (m, 2H), 3.57–3.45 (m, 2H), 3.41–3.32 (m, 1H), 3.28 (ddd, *J* = 9.2, 3.6, 1.8 Hz, 1H), 2.31 (d, *J* = 2.1 Hz, 1H), 0.83 (s, 9H), 0.01 (s, 6H) ppm. ^13^C NMR (100 MHz, chloroform-*d*) δ 138.49, 138.32, 133.03, 131.69, 128.92, 128.54, 128.47, 128.09, 128.04, 127.95, 127.84, 127.81, 87.89, 85.98, 80.44, 75.43, 75.07, 72.45, 62.14, 25.93, 18.31, −5.11, −5.34 ppm. [α]^20^_D_ = −20.3 (c 0.32, CH_2_Cl_2_); HRMS (ESI-TOF) (*m/z*): [M + Na]^+^ calculated for C_31_H_32_O_5_S_2_Na^+^, 589.2420; found, 589.2379.

*Phenyl 3*,*4-di-O-benzyl-6-O-acetyl-1-thio-β-D-glucopyranoside (**5**).* Following general process A, starting from **5a** (100 mg, 0.271 mmol), after 0.5 h of ring-opening reaction for the crude anhydro sugar, purification by silica gel flash column chromatography afforded **5** as colorless syrup (93.6 mg, 70%). Rf = 0.58 (petroleum ether/ethyl acetate 4:1); ^1^H NMR (400 MHz, chloroform-*d*) δ 7.61–7.50 (m, 2H), 7.43–7.26 (m, 13H), 4.98 (d, *J* = 11.1 Hz, 1H), 4.93–4.84 (m, 2H), 4.60 (d, *J* = 10.9 Hz, 1H), 4.52 (d, *J* = 9.7 Hz, 1H), 4.43 (dd, *J* = 11.9, 2.1 Hz, 1H), 4.23 (dd, *J* = 11.9, 5.2 Hz, 1H), 3.70–3.56 (m, 2H), 3.55–3.45 (m, 2H), 2.50 (d, *J* = 2.2 Hz, 1H), 2.08 (s, 3H) ppm. ^13^C NMR (100 MHz, chloroform-*d*) δ 170.67, 138.31, 137.64, 133.04, 131.59, 128.95, 128.57, 128.53, 128.24, 128.10, 128.04, 127.91, 127.00, 88.03, 85.91, 77.19, 75.41, 75.13, 72.72, 66.32, 63.18, 20.86 ppm. [α]^20^_D_ = −21.6 (c 0.25, CH_2_Cl_2_); HRMS (ESI-TOF) (*m/z*): [M + Na]^+^ calculated for C_28_H_30_O_6_SNa^+^, 517.1661; found, 517.1640.

*Phenyl 3*,*4-di-O-benzyl-6-O-benzoyl-1-thio-β-D-glucopyranoside (**6**).* Following general process A, starting from **6a** (100 mg, 0.232 mmol), after 0.5 h of ring-opening reaction for the crude anhydro sugar, purification by silica gel flash column chromatography afforded **6** as a colorless oil (81.5 mg, 63%). Rf = 0.36 (petroleum ether/ethyl acetate 6:1); ^1^H NMR (400 MHz, chloroform-*d*) δ 8.07–7.93 (m, 2H), 7.60 (t, *J* = 7.4 Hz, 1H), 7.54–7.05 (m, 17H), 4.96 (d, *J* = 11.0 Hz, 1H), 4.90–4.80 (m, 2H), 4.69 (dd, *J* = 12.0, 2.2 Hz, 1H), 4.61 (d, *J* = 10.8 Hz, 1H), 4.52 (d, *J* = 9.7 Hz, 1H), 4.44 (dd, *J* = 11.9, 4.7 Hz, 1H), 3.74–3.63 (m, 2H), 3.59 (t, *J* = 9.2 Hz, 1H), 3.53–3.44 (m, 1H), 2.46 (d, *J* = 2.2 Hz, 1H) ppm. ^13^C NMR (100 MHz, chloroform-*d*) δ 166.10, 138.25, 137.56, 133.34, 133.16, 131.07, 129.93, 129.77, 128.91, 128.60, 128.54, 128.42, 128.26, 128.16, 128.05, 127.98, 87.78, 85.93, 77.25, 77.01, 75.55, 75.25, 72.55, 63.37 ppm. [α]^20^_D_ = −31.3 (c 0.15, CH_2_Cl_2_); HRMS (ESI-TOF) (*m/z*): [M + Na]^+^ calculated for C_31_H_32_O_5_SNa^+^, 579.1817; found, 579.1803.

*Phenyl 3*,*4*,*6-tri-O-benzoyl-1-thio-β-D-glucopyranoside (**7**).* Following general process A, starting from **7a** (100 mg, 0.218 mmol), after 20 min of ring-opening reaction for the crude anhydro sugar, purification by silica gel flash column chromatography afforded **7** as colorless syrup (81.6 mg, 64%). Rf = 0.43 (petroleum ether/ethyl acetate 4:1); ^1^H NMR (400 MHz, chloroform-*d*) δ 8.07–7.90 (m, 6H), 7.64–7.29 (m, 12H), 7.23–7.18 (m, 2H), 5.66–5.51 (m, 2H), 4.80 (d, *J* = 9.7 Hz, 1H), 4.67 (dd, *J* = 12.2, 2.8 Hz, 1H), 4.47 (dd, *J* = 12.2, 5.8 Hz, 1H), 4.13 (ddd, *J* = 9.7, 5.7, 2.8 Hz, 1H), 3.77 (t, *J* = 9.3 Hz, 1H), 2.92–2.88 (m, 1H) ppm. ^13^C NMR (100 MHz, chloroform-*d*) δ 166.65, 166.06, 165.35, 133.51, 133.45, 133.41, 133.18, 129.91, 129.86, 129.82, 129.80, 129.67, 129.06, 129.01, 128.73, 128.53, 128.45, 128.41, 128.38, 88.30, 76.52, 76.20, 70.93, 68.93, 63.16 ppm. [α]^20^_D_ = −20.7 (c 0.058, CH_2_Cl_2_); HRMS (ESI-TOF) (*m/z*): [M + Na]^+^ calculated for C_33_H_28_O_8_SNa^+^, 607.1367; found, 607.1403.

*Phenyl 3*,*4*,*6-tri-O-acetyl-1-thio-β-D-glucopyranoside (**8**).* Following general process A, starting from **8a** (100 mg, 0.367 mmol), after 15 min of ring-opening reaction for the crude anhydro sugar, purification by silica gel flash column chromatography afforded **8** as colorless syrup (73.2 mg, 50%). Rf = 0.44 (petroleum ether/ethyl acetate 2:1); ^1^H NMR (400 MHz, chloroform-*d*) δ 7.59–7.53 (m, 2H), 7.39–7.28 (m, 3H), 5.13 (t, *J* = 9.3 Hz, 1H), 4.98 (t, *J* = 9.8 Hz, 1H), 4.57 (d, *J* = 9.7 Hz, 1H), 4.25–4.12 (m, 2H), 3.73 (ddd, *J* = 10.1, 5.0, 2.5 Hz, 1H), 3.50 (td, *J* = 9.4, 2.8 Hz, 1H), 2.53 (d, *J* = 2.9 Hz, 1H), 2.09 (s, 3H), 2.07 (s, 3H), 2.03 (s, 3H) ppm. ^13^C NMR (100 MHz, chloroform-*d*) δ 170.77, 170.61, 133.59, 130.60, 129.10, 128.72, 88.08, 75.89, 75.77, 70.31, 68.12, 62.24, 20.80, 20.76, 20.62 ppm. [α]^20^_D_ = −70.0 (c 0.05, CH_2_Cl_2_); HRMS (ESI-TOF) (*m/z*): [M + Na]^+^ calculated for C_31_H_32_O_5_SNa^+^, 421.0933; found, 421.0950.

*Phenyl 3*,*4*,*6-tri-O-ethyl-1-thio-β-D-glucopyranoside (**9**).* Following general process A, starting from **9a** (100 mg, 0.435 mmol), after 1 h of ring-opening reaction for the crude anhydro sugar, purification by silica gel flash column chromatography afforded **9** as a white solid (99.7 mg, 65%): mp 83.3–84.5 °C; Rf = 0.31 (petroleum ether/ethyl acetate 8:1); ^1^H NMR (400 MHz, chloroform-*d*) δ 7.62–7.53 (m, 2H), 7.34–7.26 (m, 3H), 4.49 (d, *J* = 9.4 Hz, 1H), 3.96–3.77 (m, 3H), 3.73 (dd, *J* = 11.0, 2.0 Hz, 1H), 3.70–3.48 (m, 4H), 4.44–4.21 (m, 4H), 2.49 (d, *J* = 2.1 Hz, 1H), 1.27–1.17 (m, 9H) ppm. ^13^C NMR (100 MHz, chloroform-*d*) δ 132.73, 128.88, 127.92, 88.04, 85.83, 79.64, 77.58, 72.25, 69.43, 68.73, 68.27, 66.97, 15.79, 15.72, 15.25 ppm. [α]^20^_D_ = −75.7 (c 0.14, CH_2_Cl_2_); HRMS (ESI-TOF) (*m/z*): [M + Na]^+^ calculated for C_31_H_32_O_5_S_2_Na^+^, 379.1555; found, 379.1558.

*Phenyl 3*,*4*,*6-tri-O-tert-butyl-dimethylsily-1-thio-β-D-glucopyranoside (**10**).* Following general process A, starting from **10a** (100 mg, 0.205 mmol), after 1 h of ring-opening reaction for the crude anhydro sugar, purification by silica gel flash column chromatography afforded **10** as a colorless oil (88 mg, 70%); Rf = 0.55 (petroleum ether/ethyl acetate 50:1); ^1^H NMR (400 MHz, chloroform-*d*) δ 7.56–7.49 (m, 2H), 7.32–7.19 (m, 3H), 4.59 (d, *J* = 8.9 Hz, 1H), 3.92 (dd, *J* = 11.3, 1.9 Hz, 1H), 3.74 (dd, *J* = 11.3, 5.6 Hz, 1H), 3.58–3.49 (m, 1H), 3.49–3.40 (m, 2H), 3.29 (ddd, *J* = 9.2, 5.7, 1.9 Hz, 1H), 2.17 (d, *J* = 2.7 Hz, 1H), 0.97–0.88 (m, 27H), 0.26–0.03 (m, 18H) ppm. ^13^C NMR (100 MHz, chloroform-*d*) δ 135.53, 130.74, 128.75, 126.78, 89.09, 81.04, 80.17, 74.40, 70.87, 62.82, 26.14, 25.96, 18.43, 18.24, −3.67, −3.85, −4.08, −4.87, −5.09, −5.33 ppm. [α]^20^_D_ = −63.8 (c 0.16, CH_2_Cl_2_); HRMS (ESI-TOF) (*m/z*): [M + Na]^+^ calculated for C_30_H_58_O_5_Si_3_SNa^+^, 637.3210; found, 637.3174.

*Phenyl 3*,*4*,*6-tri-O-p-methoxybenzyl-1-thio-β-D-glucopyranoside (**11**).* Following general process A, starting from **11a** (50 mg, 0.1 mmol), after 1 h of ring-opening reaction for the crude anhydro sugar, purification by silica gel flash column chromatography afforded **11** as a white solid (32 mg, 52%): mp 102.3–104.6 °C; Rf = 0.33 (petroleum ether/ethyl acetate 6:1); ^1^H NMR (400 MHz, chloroform-*d*) δ 7.60–7.54 (m, 2H), 7.36–7.23 (m, 7H), 7.18–7.08 (m, 2H), 6.93–6.79 (m, 6H), 4.83 (s, 2H), 4.76 (d, *J* = 10.4 Hz, 1H), 4.57 (d, *J* = 11.6 Hz, 1H), 4.54–4.46 (m, 3H), 3.83 (s, 9H), 3.79–3.67 (m, 2H), 3.60–3.43 (m, 4H), 2.40 (d, *J* = 2.1 Hz, 1H) ppm. ^13^C NMR (100 MHz, chloroform-*d*) δ 159.34, 159.32, 159.19, 132.81, 131.96, 130.66, 130.35, 130.24, 129.65, 129.37, 128.95, 128.00, 113.97, 113.82, 113.76, 88.00, 85.64, 79.47, 77.12, 74.96, 74.69, 73.09, 68.62, 55.28, 43.68, 29.71, 14.63 ppm. [α]^20^_D_ = −83.3 (c 0.09, CH_2_Cl_2_); HRMS (ESI-TOF) (*m/z*): [M + Na]^+^ calculated for C_31_H_32_O_5_S_2_Na^+^, 655.2342; found, 655.2326.

*Phenyl 3*,*4*,*6-tri-O-benzyl-1-thio-β-D-galactopyranoside (**12**).* Following general process A, starting from **12a** (50 mg, 0.12 mmol), after 1 h of ring-opening reaction for the crude anhydro sugar, purification by silica gel flash column chromatography afforded **12** as a white solid (34.2 mg, 53%): mp 89.8–90.4 °C; Rf = 0.33 (petroleum ether/ethyl acetate 4:1); ^1^H NMR (400 MHz, chloroform-*d*) δ 7.59–7.48 (m, 2H), 7.41–7.07 (m, 18H), 4.89 (d, *J* = 11.5 Hz, 1H), 4.78–4.61 (m, 2H), 4.61–4.40 (m, 4H,*H*-1, ArC*H*_2_), 4.07–3.90 (m, 2H, H-2, *H*-4), 3.66 (s, 3H, *H*-5, *H*-6a and *H*-6b), 3.48 (dd, *J* = 9.3, 2.7 Hz, 1H, *H*-3), 2.46 (d, *J* = 2.2 Hz, 1H, O*H*) ppm. ^13^C NMR (100 MHz, chloroform-*d*) δ 138.64, 137.99, 137.85, 132.60, 132.21, 128.84, 128.56, 128.46, 128.19, 127.94, 127.88, 127.85, 127.75, 127.71, 127.58, 127.47, 88.51, 83.22, 77.62, 74.41, 73.61, 73.20, 72.43, 69.07, 68.70 ppm. [α]^20^_D_ = −39.2 (c 0.13, CH_2_Cl_2_); HRMS (ESI-TOF) (*m/z*): [M + Na]^+^ calculated for C_31_H_32_O_5_S_2_Na^+^, 565.2025; found, 565.2014.

*Phenyl 3*,*4-di-O-benzyl-6-O-tert-butyl-dimethylsily-1-thio-β-D-galactopyranoside (**13**).* Following general process A, starting from **13a** (50 mg, 0.114 mmol), after 1 h of ring-opening reaction for the crude anhydro sugar, purification by silica gel flash column chromatography afforded **13** as a colorless oil (38.6 mg, 60%). Rf = 0.45 (petroleum ether/ethyl acetate 8:1); ^1^H NMR (400 MHz, chloroform-*d*) δ 7.54–7.47 (m, 2H), 7.36–7.09 (m, 13H), 4.87 (d, *J* = 11.4 Hz, 1H), 4.68 (s, 2H), 4.57 (d, *J* = 11.4 Hz, 1H), 4.48 (d, *J* = 9.6 Hz, 1H), 4.01–3.93 (m, 1H), 3.91 (d, *J* = 2.7 Hz, 1H), 3.77–3.64 (m, 2H), 3.50–3.41 (m, 2H), 2.45–2.40 (m, 1H), 0.85 (s, 9H), 0.00 (s, 6H) ppm. ^13^C NMR (100 MHz, chloroform-*d*) δ 138.85, 138.11, 132.67, 132.14, 128.83, 128.56, 128.16, 127.89, 127.76, 127.59, 127.52, 127.38, 88.52, 83.31, 79.31, 74.43, 73.08, 72.53, 69.10, 61.52, 25.93, 18.23, −5.32, −5.42 ppm. [α]^20^_D_ = +35.0 (c 0.1, CH_2_Cl_2_); HRMS (ESI-TOF) (*m/z*): [M + Na]^+^ calculated for C_32_H_42_O_5_SiSNa^+^, 589.2420; found, 589.2396.

*Phenyl 3-O-benzyl-4-O-acetyl-6-O-tert-butyl-dimethylsily-1-thio-β-D-galactopyranoside (**14**).* Following general process A, starting from **14a** (50 mg, 0.127 mmol), after 0.5 h of ring-opening reaction for the crude anhydro sugar, purification by silica gel flash column chromatography afforded **14** as a white solid (34.5 mg, 52%): mp 87.1–89.3 °C; Rf = 0.53 (petroleum ether/ethyl acetate 4:1); ^1^H NMR (400 MHz, chloroform-*d*) δ 7.55–7.48 (m, 2H), 7.32–7.21 (m, 8H), 5.56 (d, *J* = 3.0 Hz, 1H), 4.77 (d, *J* = 11.2 Hz, 1H), 4.55 (d, *J* = 9.7 Hz, 1H), 4.43 (d, *J* = 11.2 Hz, 1H), 3.73–3.65 (m, 2H), 3.64–3.54 (m, 2H), 3.46 (dd, *J* = 9.2, 3.1 Hz, 1H), 2.43 (d, *J* = 2.0 Hz, 1H), 2.04 (s, 3H), 0.84 (s, 9H), 0.00 (s, 6H) ppm. ^13^C NMR (100 MHz, chloroform-*d*) δ 170.05, 137.40, 132.64, 132.34, 128.87, 128.55, 128.27, 128.03, 127.82, 88.52, 80.45, 77.77, 71.73, 68.65, 65.98, 61.28, 25.81, 20.83, 18.21, −5.51, −5.61 ppm. [α]^20^_D_ = −110 (c 0.03, CH_2_Cl_2_); HRMS (ESI-TOF) (*m/z*): [M + Na]^+^ calculated for C_27_H_38_O_6_SiSNa^+^, 541.2056; found, 541.2047.

*Phenyl 3*,*4-di-O-benzyl-1-thio-β-D-xyloside (**15**).* Following general process A, starting from **15a** (50 mg, 0.169 mmol), after 1 h of ring-opening reaction for the crude anhydro sugar, purification by silica gel flash column chromatography afforded **15** as colorless syrup (46.3 mg, 65%). Rf = 0.53 (petroleum ether/ethyl acetate 4:1); ^1^H NMR (400 MHz, chloroform-*d*) δ 7.54–7.48 (m, 2H), 7.41–7.19 (m, 13H), 4.92 (d, *J* = 5.9 Hz, 1H, *H*-1), 4.83 (d, *J* = 11.6 Hz, 1H), 4.74 (d, *J* = 11.6 Hz, 1H), 4.63 (s, 2H), 4.29 (dd, *J* = 11.7, 3.0 Hz, 1H, *H*-5b), 3.72 (q, *J* = 6.1 Hz, 1H, *H*-2), 3.63 (t, *J* = 6.1 Hz, 1H, *H*-3), 3.59–3.45 (m, 2H, *H*-4 and *H*-5a), 3.25 (d, *J* = 6.3 Hz, 1H, O*H*) ppm. ^13^C NMR (100 MHz, chloroform-*d*) δ 138.08, 137.56, 134.18, 131.91, 128.98, 128.57, 128.53, 128.05, 127.89, 127.83, 127.57, 88.95, 79.32, 77.27, 75.92, 73.89, 72.41, 70.83, 63.55 ppm. [α]^20^_D_ = +85.0 (c 0.02, CH_2_Cl_2_); HRMS (ESI-TOF) (*m/z*): [M + Na]^+^ calculated for C_25_H_26_O_4_SNa^+^, 445.1449; found, 445.1467.

*Phenyl 2*,*3*,*3′*,*4*,*6*,*6′-hexa-O-benzyl-D-1-thio-β-lactoside (**16**)* [32]. Following general process A, starting from **16a** (100 mg, 0.118 mmol), after 1 h of ring-opening reaction for the crude anhydro sugar, purification by silica gel flash column chromatography afforded **16** as colorless syrup (64.2 mg, 56%). Rf = 0.61 (petroleum ether/ethyl acetate 3:1); ^1^H NMR (400 MHz, chloroform-*d*) δ 7.60 –7.55 (m, 2H), 7.41–7.15 (m, 33H), 5.07 (d, *J* = 11.0 Hz, 1H), 4.96 (d, *J* = 11.5 Hz, 1H), 4.85–4.73 (m, 2H), 4.73–4.62 (m, 3H), 4.58–4.47 (m, 3H), 4.48–4.37 (m, 2H), 4.34 (d, *J* = 11.7 Hz, 1H), 4.26 (d, *J* = 11.8 Hz, 1H), 3.98–3.89 (m, 2H), 3.88–3.73 (m, 3H), 3.62–3.54 (m, 1H), 3.53–3.35 (m, 6H), 2.50 (s, 1H) ppm. ^13^C NMR (100 MHz, chloroform-*d*) δ 139.00, 138.80, 138.71, 138.47, 132.97, 132.00, 128.87, 128.41, 128.38, 128.27, 128.23, 128.17, 128.13, 127.91, 127.87, 127.75, 127.67, 127.57, 127.49, 127.45, 127.35, 102.86, 87.40, 84.18, 82.51, 79.98, 79.76, 76.06, 75.37, 75.03, 74.67, 73.60, 73.46, 73.10, 73.02, 72.65, 71.59, 68.34, 68.14 ppm. [α]^20^_D_ = −46.3 (c 0.08, CH_2_Cl_2_); HRMS (ESI-TOF) (*m/z*): [M + Na]^+^ calculated for C_60_H_62_O_10_SNa^+^, 997.3961; found, 997.3990.

*4-Methylphenyl 3*,*4*,*6-tri-O-benzyl-1-thio-β-D-glucopyranoside (**17**)* [33]. Following general process A, starting from **2** (100 mg, 0.24 mmol), after 5 min of ring-opening reaction for the crude anhydro sugar, purification by silica gel flash column chromatography afforded **17** as a white solid (99.2 mg, 74%). Rf = 0.51 (petroleum ether/ethyl acetate 6:1); ^1^H NMR (400 MHz, chloroform-*d*) δ 7.50–7.41 (m, 2H), 7.40–7.15 (m, 15H), 7.10–6.97 (m, 2H), 4.91 (d, *J* = 11.2 Hz, 1H), 4.87–4.78 (m, 2H), 4.65–4.50 (m, 3H), 4.43 (d, *J* = 9.6 Hz, 1H), 3.81–3.68 (m, 2H), 3.61–3.55 (m, 2H), 3.54–3.49 (m, 1H), 3.49–3.40 (m, 1H), 2.40 (d, *J* = 2.0 Hz, 1H), 2.31 (s, 3H) ppm.

*4-Methoxyphenyl 3*,*4*,*6-tri-O-benzyl-1-thio-β-D-glucopyranoside (**18**)* [20]. Following general process A, starting from **2** (100 mg, 0.24 mmol), after 5 min of ring-opening reaction for the crude anhydro sugar, purification by silica gel flash column chromatography afforded **18** as colorless syrup (105 mg, 76%). Rf = 0.63 (petroleum ether/ethyl acetate 8:1); ^1^H NMR (400 MHz, chloroform-*d*) δ 7.60–7.46 (m, 2H), 7.41–7.29 (m, 13H), 7.25–7.13 (m, 2H), 6.82–6.70 (m, 2H), 4.96–4.80 (m, 3H), 4.68–4.53 (m, 3H), 4.39 (d, *J* = 9.6 Hz, 1H), 3.80–3.77 (m, 5H), 3.62–3.50 (m, 3H), 3.42 (dd, *J* = 9.6, 8.4 Hz, 1H), 2.43 (s, 1H) ppm.

*4-Chlorophenyl 3*,*4*,*6-tri-O-benzyl-1-thio-β-D-glucopyranoside (**19**).* Following general process A, starting from **2** (100 mg, 0.24 mmol), after 30 min of ring-opening reaction for the crude anhydro sugar, purification by silica gel flash column chromatography afforded **19** as colorless syrup (96 mg, 70%). Rf = 0.33 (petroleum ether/ethyl acetate 8:1); ^1^H NMR (400 MHz, chloroform-*d*) δ 7.56–7.49 (m, 2H), 7.42–7.18 (m, 17H), 4.94–4.79 (m, 3H), 4.64–4.54 (m, 3H), 4.49 (d, *J* = 9.6 Hz, 1H), 3.83–3.71 (m, 2H), 3.66–3.42 (m, 4H), 2.38 (d, *J* = 2.2 Hz, 1H) ppm. ^13^C NMR (100 MHz, chloroform-*d*) δ 138.38, 138.18, 137.97, 134.41, 134.32, 130.15, 129.10, 129.05, 128.81, 128.58, 128.47, 128.42, 128.00, 127.99, 127.91, 127.88, 127.70, 127.66, 87.61, 85.89, 79.35, 75.38, 75.09, 73.43, 72.44, 68.93, 38.00 ppm. [α]^20^_D_ = −115 (c 0.04, CH_2_Cl_2_); HRMS (ESI-TOF) (*m/z*): [M + Na]^+^ calculated for C_33_H_33_O_5_ClSNa^+^, 599.1635; found, 599.1611.

*4-Aminophenyl 3*,*4*,*6-tri-O-benzyl-1-thio-β-D-glucopyranoside (**20**).* Following general process A, starting from **2** (50 mg, 0.12 mmol), after 30 min of ring-opening reaction for the crude anhydro sugar, purification by silica gel flash column chromatography afforded **20** as colorless syrup (45 mg, 68%). Rf = 0.35 (petroleum ether/ethyl acetate 2:1); ^1^H NMR (600 MHz, CDCl_3_) δ 7.41–7.25 (m, 15H), 7.24–7.20 (m, 2H), 6.59–6.54 (m, 2H), 4.94 (d, *J* = 11.2 Hz, 1H), 4.88–4.81 (m, 2H), 4.66–4.54 (m, 3H), 4.33 (d, *J* = 9.6 Hz, 1H), 4.15 (q, *J* = 7.1 Hz, 1H), 3.82–3.74 (m, 2H), 3.64–3.54 (m, 2H), 3.42 (t, *J* = 9.0 Hz, 1H), 2.43 (s, 1H) ppm. ^13^C NMR (100 MHz, chloroform-*d*) δ 147.25, 138.57, 136.38, 128.48, 128.40, 128.33, 127.98, 127.76, 127.64, 127.49, 115.33, 88.26, 85.91, 79.46, 75.26, 75.06, 73.43, 72.21, 69.04, 29.71, 29.33 ppm. [α]^20^_D_ = +61.5 (c 0.026, CH_2_Cl_2_); HRMS (ESI-TOF) (*m/z*): [M + Na]^+^ calculated for C_31_H_32_O_5_S_2_Na^+^, 580.2130; found, 580.2120.

*Thiophen-2-ylthio 3*,*4*,*6-tri-O-benzyl-1-thio-β-D-glucopyranoside (**21**).* Following general process A, starting from **2** (50 mg, 0.12 mol), after 30 min of ring-opening reaction for the crude anhydro sugar, purification by silica gel flash column chromatography afforded **21** as a red solid (41 mg, 63%): mp 90.9–92.1 °C; Rf = 0.52 (petroleum ether/ethyl acetate 5:1); ^1^H NMR (400 MHz, chloroform-*d*) δ 7.47–7.09 (m, 17H), 6.98 (dd, *J* = 5.5, 3.5 Hz, 1H), 4.92–4.76 (m, 3H), 4.70–4.51 (m, 3H), 4.30 (d, *J* = 9.4 Hz, 1H), 3.82–3.70 (m, 2H), 3.62–3.46 (m, 3H), 3.42 (ddd, *J* = 9.0, 6.2, 2.7 Hz, 1H), 2.36 (d, *J* = 2.4 Hz, 1H) ppm. ^13^C NMR (100 MHz, chloroform-*d*) δ 138.39, 138.04, 136.17, 131.08, 128.56, 128.43, 128.34, 128.00, 127.96, 127.88, 127.81, 127.63, 127.53, 87.53, 85.80, 79.69, 75.39, 75.06, 73.50, 71.86, 68.87, 29.39 ppm. [α]^20^_D_ = −46.6 (c 0.058, CH_2_Cl_2_); HRMS (ESI-TOF) (*m/z*): [M + Na]^+^ calculated for C_31_H_32_O_5_S_2_Na^+^, 571.1589; found, 571.1602.

*Benzothiazol-2-yl 3*,*4*,*6-tri-O-benzyl-1-thio-β-D-glucopyranoside (**22**).* Following general process A, starting from **2** (50 mg, 0.12 mmol), after 30 min of ring-opening reaction for the crude anhydro sugar, purification by silica gel flash column chromatography afforded **22** as a white solid (51.3 mg, 72%): mp 119.9–123.0 °C; Rf = 0.36 (petroleum ether/ethyl acetate 6:1); ^1^H NMR (400 MHz, chloroform-*d*) δ 7.94 (d, *J* = 8.1 Hz, 1H), 7.72 (d, *J* = 8.0 Hz, 1H), 7.45 (t, *J* = 7.7 Hz, 1H), 7.39–7.23 (m, 14H), 7.23–7.17 (m, 2H), 5.07 (d, *J* = 9.5 Hz, 1H), 4.99–4.80 (m, 3H), 4.67–4.47 (m, 3H), 3.86–3.63 (m, 6H), 3.13 (d, *J* = 3.1 Hz, 1H) ppm. ^13^C NMR (100 MHz, chloroform-*d*) δ 211.54, 152.68, 138.40, 138.11, 138.00, 128.56, 128.45, 128.36, 128.01, 127.95, 127.89, 127.85, 127.76, 127.61, 126.31, 125.06, 122.43, 121.02, 86.49, 85.95, 79.94, 75.49, 75.10, 73.50, 68.65, 29.34, 14.14 ppm. [α]^20^_D_ = −64.3 (c 0.07, CH_2_Cl_2_); HRMS (ESI-TOF) (*m/z*): [M + Na]^+^ calculated for C_34_H_33_NO_5_S_2_Na^+^, 622.1698; found, 622.1707.

*Phenyl 3*,*4*,*6-tri-O-benzyl-1-seleno-β-D-glucopyranoside (**26**)* [34]. Following general process A, starting from **2** (100 mg, 0.24 mmol), after 12 h of ring-opening reaction for the crude anhydro sugar, purification by silica gel flash column chromatography afforded **26** as a white solid (85.5 mg, 61%). Rf = 0.43 (petroleum ether/ethyl acetate 8:1); ^1^H NMR (400 MHz, chloroform-*d*) δ 7.72–7.66 (m, 2H), 7.41–7.19 (m, 18H), 4.93 (d, *J* = 11.2 Hz, 1H), 4.89–4.82 (m, 2H), 4.75 (d, *J* = 9.7 Hz, 1H), 4.67–4.54 (m, 3H), 3.85–3.74 (m, 2H), 3.67–3.57 (m, 2H), 3.56–3.47 (m, 2H) ppm.

*Phenyl 3*,*4*,*6-tri-O-benzyl-2-O-picolyl-β-D-1-thio-glucopyranoside (**28**).* Following the general process B, starting from **2** (100 mg, 0.24 mmol), after 2 h of ring-opening reaction for the crude anhydro sugar and then reaction with PicBr•HBr (2.0 equiv) for 1 h in the presence of sodium hydride (6 equiv), the residue was purified by column chromatography on silica gel (ethyl acetate-hexane gradient elution) to afford **28** as a colorless syrup (101 mg, 66%). Rf = 0.33 (petroleum ether/ethyl acetate 4:1); ^1^H NMR (400 MHz, chloroform-*d*) δ 8.58–8.50 (m, 1H), 7.65 (td, *J* = 7.7, 1.9 Hz, 1H), 7.58–7.49 (m, 3H), 7.40–7.09 (m, 19H), 5.04 (d, *J* = 12.4 Hz, 1H), 4.96–4.75 (m, 4H), 4.69 (d, *J* = 9.6 Hz, 1H), 4.64–4.49 (m, 3H), 3.84–3.69 (m, 3H), 3.65 (t, *J* = 9.3 Hz, 1H), 3.55 (t, *J* = 9.1 Hz, 2H) ppm. ^13^C NMR (100 MHz, chloroform-*d*) δ 158.30, 149.00, 138.29, 138.20, 138.06, 136.52, 133.54, 132.04, 128.90, 128.44, 128.41, 128.35, 128.00, 127.92, 127.82, 127.70, 127.65, 127.56, 127.50, 122.35, 121.65, 87.19, 86.55, 81.32, 79.13, 75.93, 75.81, 75.06, 73.43, 69.04 ppm. [α]^20^_D_ = −45 (c 0.04, CH_2_Cl_2_); HRMS (ESI-TOF) (*m/z*): [M + Na]^+^ calculated for C_39_H_39_O_5_NSNa^+^, 656.2447; found, 656.2416.

*Phenyl 3*,*4*,*6-tri-O-benzyl-2-O-picolyl-β-D-1-thio-galactopyranoside (**29**).* Following the general process B, starting from **12a** (100 mg, 0.24 mmol), after 2 h of ring-opening reaction for the crude anhydro sugar, and then reaction with PicBr•HBr (2.0 equiv) for 1 h in the presence of sodium hydride (6 equiv), the residue was purified by column chromatography on silica gel (ethyl acetate-hexane gradient elution) to afford **29** as a colorless syrup (73 mg, 48%). Rf = 0.35 (petroleum ether/ethyl acetate 4:1); ^1^H NMR (400 MHz, chloroform-*d*) δ 8.53 (d, *J* = 5.0 Hz, 1H), 7.62 (t, *J* = 7.6 Hz, 1H), 7.56–7.43 (m, 3H), 7.41–7.09 (m, 19H), 4.98–4.87 (m, 3H), 4.75–4.63 (m, 3H), 4.58 (d, *J* = 11.5 Hz, 1H), 4.51–4.37 (m, 2H), 4.01–3.90 (m, 2H), 3.72–3.57 (m, 4H) ppm. ^13^C NMR (100 MHz, chloroform-*d*) δ 158.67, 148.91, 138.75, 138.14, 137.89, 136.40, 133.94, 131.56, 128.80, 128.46, 128.39, 128.22, 127.96, 127.86, 127.84, 127.64, 127.49, 127.11, 122.24, 121.79, 87.52, 83.91, 78.04, 77.40, 77.34, 77.08, 76.76, 76.28, 74.49, 73.62, 73.48, 72.57, 68.79 ppm. [α]^20^_D_ = +9.1 (c 0.33, CH_2_Cl_2_); HRMS (ESI-TOF) (*m/z*): [M + Na]^+^ calculated for C_39_H_39_O_5_NSNa^+^, 656.2447; found, 656.2433.

*Phenyl 3*,*4*,*6-Tri-O-benzyl-2-O-(phenylmethoxy)methyl-β-D-1-thio-glucopyranoside (**30**)* [21]. Following the general process B, starting from **2** (100 mg, 0.24 mmol), after 2 h of ring-opening reaction for the crude anhydro sugar and then reaction with BOMCl (2.0 equiv) for 3 h in the presence of sodium hydride (6 equiv), the residue was purified by column chromatography on silica gel (ethyl acetate-hexane gradient elution) to afford **30** as a white solid (100.3 mg, 63%). Rf = 0.53 (petroleum ether/ethyl acetate 6:1); ^1^H NMR (400 MHz, CDCl_3_) δ 7.59–7.49 (m, 2H), 7.39–7.12 (m, 23H), 5.06 (d, *J* = 6.4 Hz, 1H), 4.94 (d, *J* = 6.4 Hz, 1H), 4.92 –4.83 (m, 3H), 4.79 (d, *J* = 10.8 Hz, 1H), 4.69–4.58 (m, 3H), 4.58–4.49 (m, 2H), 3.77 (dd, *J* = 10.9, 2.0 Hz, 1H), 3.74–3.61 (m, 4H), 3.52 (m, 1H) ppm.

*Phenyl 3*,*4*,*6-Tri-O-benzyl-2-O-(cyanomethyl)-β-D-1-thio-glucopyranoside (**31**)* [18]. Following the general process B, starting from **2** (100 mg, 0.24 mmol), after 2 h of ring-opening reaction for the crude anhydro sugar and then reaction with bromoacetonitrile (2.5 equiv) for 2 h in the presence of sodium hydride (6 equiv), the residue was purified by column chromatography on silica gel (ethyl acetate-hexane gradient elution) to afford **31** as a white solid (76.8 mg, 55%). Rf = 0.53 (petroleum ether/ethyl acetate 6:1); ^1^H NMR (400 MHz, CDCl_3_) δ 7.61–7.52 (m, 2H), 7.41–7.22 (m, 16H), 7.19 (dd, *J* = 7.2, 2.3 Hz, 2H), 4.90–4.82 (m, 2H), 4.80 (d, *J* = 10.9 Hz, 1H), 4.64–4.56 (m, 2H), 4.55 (s, 1H), 4.52 (t, *J* = 1.8 Hz, 1H), 4.51–4.39 (m, 2H), 3.77 (dd, *J* = 10.9, 2.1 Hz, 1H), 3.72 (dd, *J* = 10.9, 4.3 Hz, 1H), 3.67–3.62 (m, 2H), 3.47 (m, 1H), 3.35 (m, 1H) ppm.

*Phenyl 3*,*4*,*6-Tri-O-benzyl-2-O-(2-cyanobenzyl)-β-D-1-thio-glucopyranoside (**32**)* [22]. Following the general process B, starting from **2** (100 mg, 0.24 mmol), after 2 h of ring-opening reaction for the crude anhydro sugar and then reaction with 2-cyanobenzyl bromide (2.0 equiv) for 1 h in the presence of sodium hydride (6 equiv), the residue was purified by column chromatography on silica gel (ethyl acetate-hexane gradient elution) to afford **32** as a colorless oil (107.4 mg, 68%). Rf = 0.31 (petroleum ether/ethyl acetate 6:1); ^1^H NMR (400 MHz, CDCl_3_) δ 7.73–7.53 (m, 6H), 7.43–7.19 (m, 18H), 5.11 (d, *J* = 12.5 Hz, 1H), 5.03 (d, *J* = 12.5 Hz, 1H), 4.91–4.79 (m, 3H), 4.73–4.56 (m, 4H), 3.83 (dd, *J* = 10.9, 2.0 Hz, 1H), 3.78 (d, *J* = 4.4 Hz, 1H), 3.77–3.66 (m, 2H), 3.60–3.51 (m, 2H) ppm.

*Phenyl 3*,*4*,*6-Tri-O-benzyl-2-O-benzoyl-β-D-1-thio-glucopyranoside (**33**)* [35]. Following the general process B, starting from **2** (100 mg, 0.24 mmol), after 2 h of ring-opening reaction for the crude anhydro sugar and then reaction with BzCl (3.0 equiv) for 2 h in the presence of sodium hydride (6 equiv), the residue was purified by column chromatography on silica gel (ethyl acetate-hexane gradient elution) to afford **33** as a white solid (97.8 mg, 63%). Rf = 0.61 (petroleum ether/ethyl acetate 6:1); ^1^H NMR (400 MHz, CDCl_3_) δ 8.11–8.03 (m, 2H), 7.60 (d, *J* = 7.4 Hz, 1H), 7.54–7.44 (m, 4H), 7.43–7.20 (m, 13H), 7.18–7.11 (m, 5H), 5.31 (dd, *J* = 10.0, 9.0 Hz, 1H), 4.87–4.78 (m, 2H), 4.75 (d, *J* = 11.0 Hz, 1H), 4.70–4.55 (m, 4H), 3.91–3.82 (m, 2H), 3.81–3.74 (m, 2H), 3.64 (m, 1H) ppm.

*Phenyl 3*,*4*,*6-Tri-O-benzyl-2-O-pivaloyl-β-D-1-thio-glucopyranoside (**34**)* [36]. Following the general process B, starting from **2** (100 mg, 0.24 mmol), after 2 h of ring-opening reaction for the crude anhydro sugar and then reaction with PivCl (3.0 equiv) for 2 h in the presence of sodium hydride (6 equiv), the residue was purified by column chromatography on silica gel (ethyl acetate-hexane gradient elution) to afford **34** as a white solid (78.3 mg, 52%). Rf = 0.73 (petroleum ether/ethyl acetate 8:1); ^1^H NMR (400 MHz, CDCl_3_) δ 7.58–7.49 (m, 2H), 7.41–7.22 (m, 16H), 7.19 (dd, *J* = 7.2, 2.4 Hz, 2H), 5.13 (t, 1H), 4.84–4.76 (m, 2H), 4.75–4.53 (m, 5H), 3.81 (dd, 1 H, J = 1.5, 11.0 Hz, 1H), 3.79–3.67 (m, 3H), 3.59 (m, 1H), 1.26 (s, 9H) ppm.

*Methyl 2*,*3*,*4-tri-O-benzoyl-6-O-(3*,*4*,*6-tri-O-benzyl-2-O-picolyl-β-D-glucopyranosyl)-α-D-glucopyranoside (**35**).* Following the general process C, the glycosylation between **28** (100 mg, 0.16 mmol, 1.3 equiv) and methyl 2,3,4-tri-*O*-benzoyl-α-D-glucopyranoside (1.0 equiv) led to **30**. Purification by silica gel flash column chromatography afforded **35** as a colorless syrup (87.5 mg, 70%, β-only). Rf = 0.26 (petroleum ether/ethyl acetate 2:1); ^1^H NMR (400 MHz, chloroform-*d*) δ 8.53 (d, *J* = 4.9 Hz, 1H), 8.03–7.76 (m, 7H), 7.66–7.58 (m, 1H), 7.55–7.46 (m, 3H), 7.45–7.20 (m, 19H), 7.17–7.07 (m, 3H), 6.14 (t, *J* = 9.8 Hz, 1H), 5.42 (t, *J* = 9.9 Hz, 1H), 5.24–5.16 (m, 2H), 5.11 (d, *J* = 3.6 Hz, 1H), 4.95–4.83 (m, 2H), 4.82–4.73 (m, 2H), 4.56–4.46 (m, 3H), 4.44 (d, *J* = 12.3 Hz, 1H), 4.34 (td, *J* = 8.7, 8.2, 4.1 Hz, 1H), 4.10 (dd, *J* = 10.9, 2.1 Hz, 1H), 3.77 (dd, *J* = 11.1, 7.8 Hz, 1H), 3.73–3.57 (m, 4H), 3.53–3.41 (m, 2H), 3.34 (s, 3H) ppm. ^13^C NMR (100 MHz, chloroform-*d*) δ 165.81, 165.74, 165.50, 158.78, 148.95, 138.50, 138.14, 138.09, 136.46, 133.42, 133.33, 133.05, 129.93, 129.90, 129.66, 129.28, 129.11, 128.90, 128.41, 128.34, 128.25, 127.98, 127.94, 127.75, 127.58, 122.18, 121.45, 103.94, 96.68, 84.47, 82.78, 77.65, 75.68, 75.40, 75.00, 74.95, 73.45, 72.11, 70.49, 69.91, 68.97, 68.61, 55.52 ppm. [α] ^20^_D_ = +40.9 (c 0.22, CH_2_Cl_2_); HRMS (ESI-TOF) (*m/z*): [M + Na]^+^ calculated for C_61_H_59_O_14_NNa^+^, 1052.3833 ; found, 1052.3814.

*Methyl 2*,*3*,*4-tri-O-benzyl-6-O-(3*,*4*,*6-tri-O-benzyl-2-O-picolyl**-β-D-glucopyranosyl)-α-D-glucopyranoside (**36**)* [37]. Following the general process C, the glycosylation between **28** (100 mg, 0.16 mmol, 1.3 equiv) and methyl 2,3,4-tri-*O*-benzyl-α-D-glucopyranoside (1.0 equiv) led to **36**. Purification by silica gel flash column chromatography afforded **36** as a colorless syrup (97.7 mg, 82%, β-only). Rf = 0.46 (petroleum ether/ethyl acetate 2:1); ^1^H NMR (400 MHz, chloroform-*d*) δ 8.44 (d, *J* = 4.8 Hz, 1H), 7.48–7.20 (m, 28H), 7.19–7.11 (m, 4H), 7.02 (t, *J* = 6.1 Hz, 1H), 5.13 (d, *J* = 12.9 Hz, 1H), 4.96–4.85 (m, 3H), 4.84–4.70 (m, 4H), 4.68–4.59 (m, 2H), 4.58–4.51 (m, 4H), 4.47 (d, *J* = 11.0 Hz, 1H), 4.39 (d, *J* = 7.8 Hz, 1H), 4.17 (d, *J* = 10.7 Hz, 1H), 3.95 (t, *J* = 9.3 Hz, 1H), 3.78 (dt, *J* = 14.3, 7.1 Hz, 1H), 3.72–3.61 (m, 4H), 3.60–3.40 (m, 5H), 3.31 (s, 3H) ppm.

*Methyl 2*,*3*,*6-tri-O-benzyl-4-O-(3*,*4*,*6-tri-O-benzyl-2-O-picolyl-β-D-glucopyranosyl)-α-D-glucopyranoside (**37**)* [37]. Following the general process C, the glycosylation between **28** (100 mg, 0.16 mmol, 1.3 equiv) and methyl 2,3,6-tri-*O*-benzyl-α-D-glucopyranoside (1.0 equiv) led to **37**. Purification by silica gel flash column chromatography afforded **37** as a colorless syrup (59.6 mg, 50%, β-only). Rf = 0.48 (petroleum ether/ethyl acetate 2:1); ^1^H NMR (400 MHz, chloroform-*d*) δ 8.50 (d, *J* = 4.8 Hz, 1H), 7.54 (t, *J* = 7.7 Hz, 1H), 7.45–7.06 (m, 32H), 5.07 (d, *J* = 11.3 Hz, 1H), 4.96 (d, *J* = 13.5 Hz, 1H), 4.90–4.71 (m, 6H), 4.64–4.52 (m, 4H), 4.48–4.34 (m, 4H), 3.96 (t, *J* = 9.5 Hz, 1H), 3.87–3.77 (m, 2H), 3.72 (d, *J* = 10.9 Hz, 1H), 3.65–3.57 (m, 2H), 3.48 (m, 5H), 3.34 (s, 3H), 3.31–3.27 (m, 1H) ppm.

*Methyl 2*,*4*,*6-tri-O-benzyl-3-O-(3*,*4*,*6-tri-O-benzyl-2-O-picolyl-β-D-glucopyranosyl)-α-D-glucopyranoside (**38**)* [37]. Following the general process C, the glycosylation between **28** (100 mg, 0.16 mmol, 1.3 equiv) and methyl 2,4,6-tri-*O*-benzyl-α-D-glucopyranoside (1.0 equiv) led to **38**. Purification by silica gel flash column chromatography afforded **38** as a colorless syrup (95.4 mg, 80%, β-only). Rf = 0.65 (petroleum ether/ethyl acetate 2:1); ^1^H NMR (400 MHz, chloroform-*d*) δ 8.56 (d, *J* = 4.9 Hz, 1H), 7.56–7.47 (m, 2H), 7.40–7.04 (m, 31H), 5.29 (d, *J* = 13.5 Hz, 1H), 5.14–4.95 (m, 4H), 4.89 (d, *J* = 10.9 Hz, 1H), 4.83 (d, *J* = 10.6 Hz, 1H), 4.70–4.53 (m, 3H), 4.53–4.33 (m, 7H), 3.81–3.64 (m, 6H), 3.63–3.48 (m, 4H), 3.45 (d, *J* = 9.4 Hz, 1H), 3.30 (s, 3H) ppm.

*Methyl 3*,*4*,*6-tri-O-benzyl-2-O-(3*,*4*,*6-tri-O-benzyl-2-O-picolyl-β-D-glucopyranosyl)-α-D-glucopyranoside (**39**)* [37]. Following the general process C, the glycosylation between **28** (100 mg, 0.16 mmol, 1.3 equiv) and methyl 3,4,6-tri-*O*-benzyl-α-D-glucopyranoside led to **39**. Purification by silica gel flash column chromatography afforded **39** as a colorless syrup (91.8 mg, 77%, β-only). Rf = 0.55 (petroleum ether/ethyl acetate 2:1); ^1^H NMR (400 MHz, chloroform-*d*) δ 8.42 (d, *J* = 5.0 Hz, 1H), 7.45–7.10 (m, 30H), 7.11–6.96 (m, 3H), 5.18 (d, *J* = 13.4 Hz, 1H), 5.03–4.85 (m, 3H), 4.85–4.70 (m, 4H), 4.67–4.56 (m, 3H), 4.55–4.40 (m, 4H), 4.01 (t, *J* = 9.2 Hz, 1H), 3.89–3.61 (m, 10H), 3.58 (t, *J* = 7.4 Hz, 1H), 3.43 (m, 1H), 3.39 (s, 3H) ppm.

*1*,*2:5*,*6-Di-O-isopropylidine-3-O-(3*,*4*,*6-tri-O-benzyl-2-O-picolyl-β-D-glucopyranosyl)-α-D-glucofuranose (**40**)* [37]. Following the general process C, the glycosylation between **28** (100 mg, 0.16 mmol, 1.3 equiv) and 1,2:5,6-bis-*O*-(1-methylethylidene)-α-D-glucofuranose (1.0 equiv) led to **40**. Purification by silica gel flash column chromatography afforded **40** as colorless syrup (49.2 mg, 52%, β-only). Rf = 0.35 (petroleum ether/ethyl acetate 2:1); ^1^H NMR (400 MHz, chloroform-*d*) δ 8.56 (d, *J* = 4.9 Hz, 1H), 7.67–7.56 (m, 1H), 7.43–7.22 (m, 14H), 7.21–7.08 (m, 3H), 5.81 (d, *J* = 3.7 Hz, 1H), 4.93–4.86 (m, 2H), 4.85–4.75 (m, 3H), 4.66–4.47 (m, 5H), 4.43 (q, *J* = 5.9 Hz, 1H), 4.38–4.28 (m, 2H), 4.12–4.01 (m, 2H), 3.78–3.58 (m, 4H), 3.47–3.38 (m, 2H), 1.48 (s, 3H), 1.41 (s, 3H), 1.31 (s, 3H), 1.25 (s, 3H) ppm.

*Methyl 2*,*3*,*4-tri-O-benzyl-6-O-(3*,*4*,*6-tri-O-benzyl-2-O-picolyl-β-D-galactopyranosyl)-α-D-glucopyranoside (**41**)* [37]. Following the general process C, the glycosylation between **29** (100 mg, 0.16 mmol, 1.3 equiv) and methyl 2,3,4-tri-*O*-benzyl-α-D-glucopyranoside (1.0 equiv) led to **41**. Purification by silica gel flash column chromatography afforded **41** as a colorless syrup (102.5 mg, 86%, β-only). Rf = 0.36 (petroleum ether/ethyl acetate 2:1); ^1^H NMR (400 MHz, chloroform-*d*) δ 8.43 (d, *J* = 4.8 Hz, 1H), 7.48–7.39 (m, 2H), 7.39–7.08 (m, 30H), 7.04 (q, *J* = 4.7 Hz, 1H), 5.09 (d, *J* = 13.2 Hz, 1H), 4.98–4.87 (m, 3H), 4.80–4.38 (m, 11H), 4.35 (d, *J* = 7.7 Hz, 1H), 4.13 (d, *J* = 10.8 Hz, 1H), 3.97–3.84 (m, 3H), 3.79 (dd, *J* = 10.5, 5.0 Hz, 1H), 3.52 (m, 7H), 3.28 (s, 3H) ppm.

## Figures and Tables

**Figure 1 molecules-27-05980-f001:**
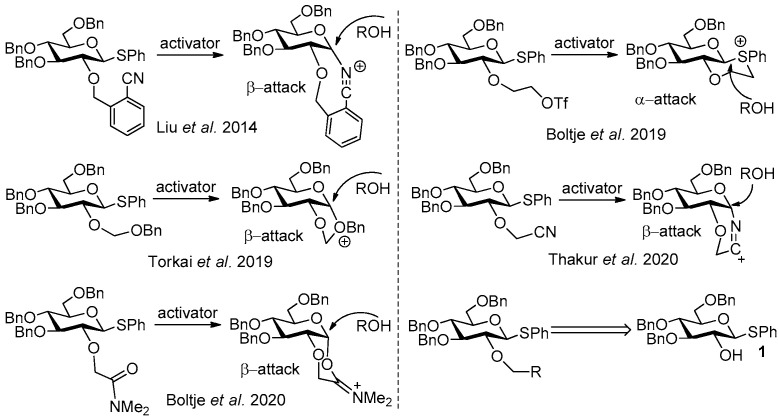
Using NGP from 2-ether groups to control stereoselectivity.

**Figure 2 molecules-27-05980-f002:**
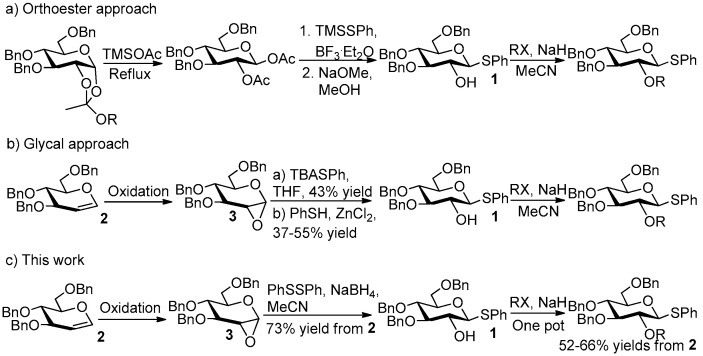
Comparison of this method with previous methods.

**Figure 3 molecules-27-05980-f003:**
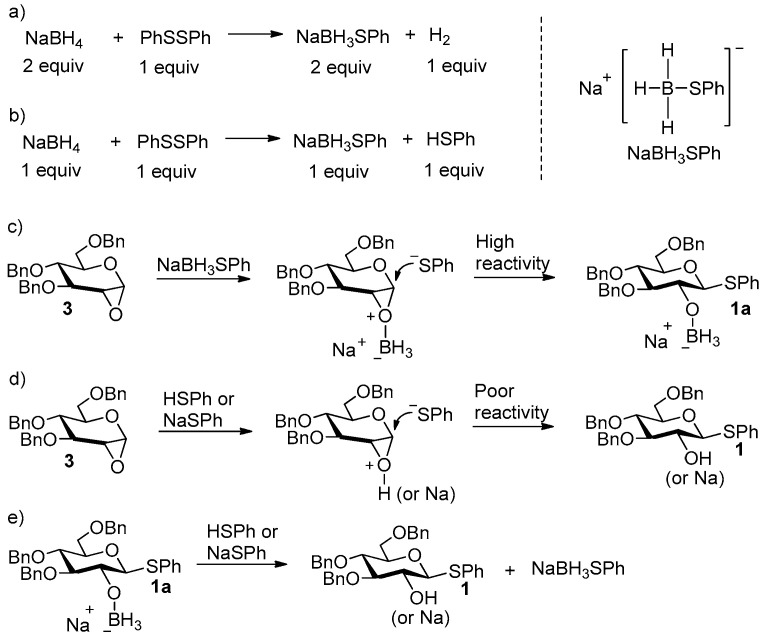
Proposed reaction mechanism.

**Figure 4 molecules-27-05980-f004:**
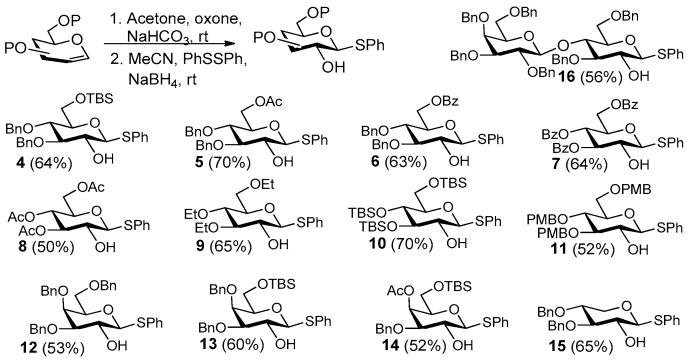
Synthesis of 1-thiophenyl glycosides with 2-OH starting from corresponding glycals.

**Figure 5 molecules-27-05980-f005:**
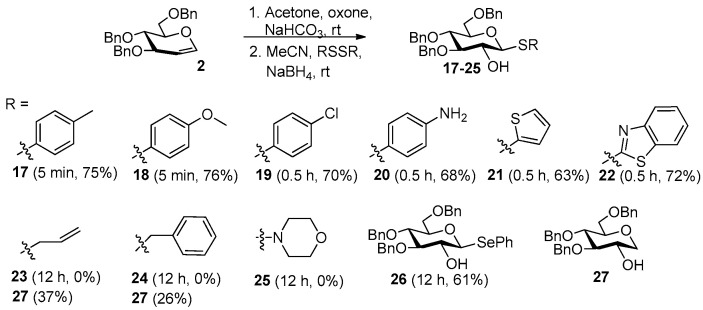
Evaluation of reactions between various disulfides and glucal 2.

**Figure 6 molecules-27-05980-f006:**
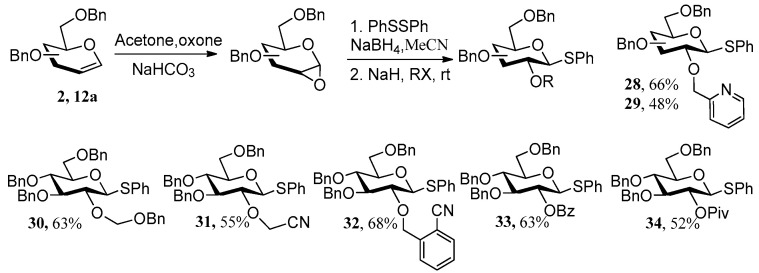
One-pot synthesis of various thioglycoside donors.

**Figure 7 molecules-27-05980-f007:**
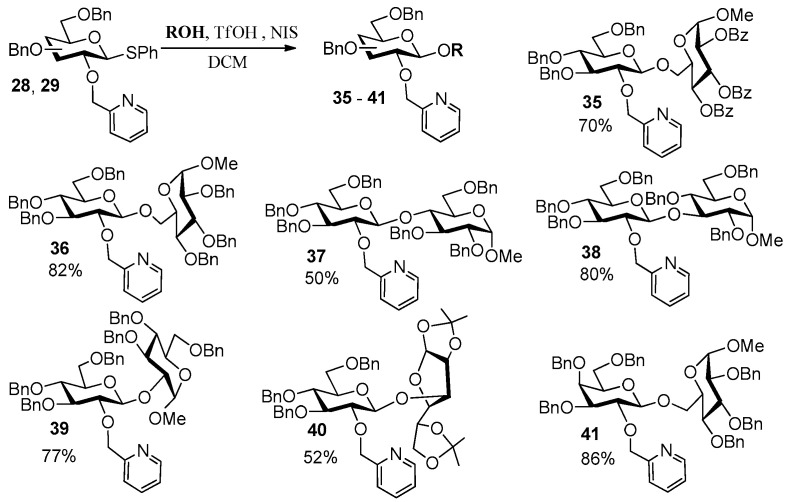
Application of this method in glycosylation.

**Table 1 molecules-27-05980-t001:** Comparison of results by variation of reaction conditions ^a^.

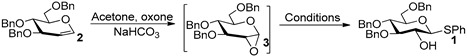
Entry	NaBH_4_ (equiv)	PhSSPh (equiv)	Reaction Conditions(equiv)	Yields (%)
1	1.5	0.7	ACN, **rt**/0 °C, **1**/4 h	**72**/75
2	1.5	0.7	ACN, rt, 1 h	**73**^b^/70 ^c^
3	**1****.0**/0.7	0.7	ACN, rt, 1 h	**68**/60
4	**1.0**/2.0	0.5/1.0	ACN, rt, 1 h	**55**/65
5	1.2	**0.6**/0.8	ACN, rt, 1 h	**65**/69
6	1.5	0.7	**Acetone**/DMF, rt, 1 h	**18**/<5
7	1.5	0.7	**MeOH**/DCM, rt, 1 h	**21**/-
8	-	-	ACN, NaSPh (1.5), rt, 36 h	43
9	0.1	0.1	ACN, NaSPh (1.2), rt, 4 h	55
10	0.3	0.3	ACN, NaSPh (0.8), rt, 4 h	72
11	0.5	0.5	ACN, NaSPh (0.4), rt, 4 h	65

^a^ Reagents and conditions: substrate **2** (0.1 mmol), solvents (1 mL), yields based on **2**. ^b^ Treatment of PhSSPh with NaBH_4_ in acetonitrile at 50 °C for 1 h, then cooling to rt, and adding crude substrate **3**. ^c^ Large scale.

## Data Availability

Not applicable.

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
