# Peer review of "Efficient Synthesis of 2-OH Thioglycosides from Glycals Based on the Reduction of Aryl Disulfides by NaBH4"

_molecules, 2022, doi:10.3390/molecules27185980_

Round 1

Reviewer 1 Report

In this study, the authors reported an one-pot method to efficiently synthesize 2-OH thioaryl glycosides starting from corresponding per-protected glycals and disulfides, via 1,2-anhydro sugars intermediate. This method has been further used to synthesize glycosyl donors with both “armed” and “NGP (neighboring group participation)” effects’ groups on C2 position.

I find the work well conducted and the results worth being published. Some points could be improved to increase the overall quality of the manuscript. I list them here below:

1, P1 line 17: The first sentence in introduction, please reconsider "a major challenge" or "an important research topic".

2, P1 line 21: ---produce a product in the alfa or beta configuration, which will lead to --reduced yield and difficult purification----, may be use "and"

3, P1 line 26: --which leads to the beta-configuration----  it is not precise, may be can use "1,2-trans" 

4, P2 figure 2: ZnCl2 should be ZnCl2

5, P3 line 95: (1, 42%, 36 h) ,compound number should be bold.

6, P4 compounds numbers 4b-11b, 12b - 14b are not in accordance with the numbers in figure 4.

7, P7 line 198: In general process B,  NaH is indispensable or not?

8, P14 line 595: the type for authors' name is not the same. H. Dong should be Dong, H.

9, P 15 line 619 and 646: Angew. Chem Int. Ed. should be Angew. Chem. Int. Ed.

Author Response

1, P1 line 17: The first sentence in introduction, please reconsider "a major challenge" or "an important research topic".

Answer: Good suggestion. We changed  "a major challenge" as "an important research topic"

2, P1 line 21: ---produce a product in the alfa or beta configuration, which will lead to --reduced yield and difficult purification----, may be use "and"

Answer: Here "produce a product", so using "or" is right. When "produce a mixtue of two products", using "and" is right. 

3, P1 line 26: --which leads to the beta-configuration----  it is not precise, may be can use "1,2-trans" 

Answer: The reviewer is right. We have corrected it.

4, P2 figure 2: ZnCl2 should be ZnCl2.

Answer: The reviewer is right. We have corrected it. 

5, P3 line 95: (1, 42%, 36 h) ,compound number should be bold.

Answer: The reviewer is right. We have corrected it.

6, P4 compounds numbers 4b-11b, 12b - 14b are not in accordance with the numbers in figure 4.

Answer: The reviewer is right. We have corrected them.

7, P7 line 198: In general process B,  NaH is indispensable or not?
Answer: It is a cloudy solution. Most likely NaH is not completely soluble.

8, P14 line 595: the type for authors' name is not the same. H. Dong should be Dong, H.

Answer: The reviewer is right. We have corrected it.

9, P 15 line 619 and 646: Angew. Chem Int. Ed. should be Angew. Chem. Int. Ed.

Answer: The reviewer is right. We have corrected it.

Reviewer 2 Report

In this manuscript, Dong and co-workers report an efficient method for synthesis of 2-OH thioaryl glycosides by reduction of aryl disulfides with NaBH4. Furthermore, the C2-protected thioglycoside donors were also prepared in a one-pot reaction, and then used for glycosylation. Therefore, the results obtained in this study will have a sufficient impact for the readers of <molecules>, and I recommend publication of this manuscript after revision of the following points.

1.    After checking the experimental procedure, it was found that 35~90 min should be needed for synthesis of 1 from compound 2. Thus, in the sentence (line 47~ 48) “As a result, 73% yield of 1 could be efficiently prepared from 2 over a 30-min reaction…” should be changed to “As a result, 73% yield of 1 could be efficiently prepared from 2 within 90 min…” or others.

2.    In Figure 2 and Figure 6, “ZnCl2” and “NaBH4” should be changed to “ZnCl2” and “NaBH4” respectively.

3.    In the whole text, the letter “D” in “D-glucopyranosides” should be changed to the small caps “d”.

4.    In line 464, the color of ref [4d] should be corrected.

5.    In SI, please check the grammar of the sentence “Synthesized in light of the reported reference”.

Author Response

  1. After checking the experimental procedure, it was found that 35~90 min should be needed for synthesis of 1 from compound 2. Thus, in the sentence (line 47~ 48) “As a result, 73% yield of 1 could be efficiently prepared from 2 over a 30-min reaction…” should be changed to “As a result, 73% yield of 1 could be efficiently prepared from 2 within 90 min…” or others.
    Answer:The reviewer is right. We have corrected this sentence.
  2. In Figure 2 and Figure 6, “ZnCl2” and “NaBH4” should be changed to “ZnCl2” and “NaBH4” respectively.
    Answer:We have corrected these Figures.
  3. In the whole text, the letter “D” in “D-glucopyranosides” should be changed to the small caps “d”.
    Answer:We have corrected them.
  4. In line 464, the color of ref [4d] should be corrected.
    Answer:We have corrected it.
  5. In SI, please check the grammar of the sentence “Synthesized in light of the reported reference”.
    Answer:what we used here is an incomplete sentence. The complete sentence is "The compound is synthesized in light of the reported reference."

Reviewer 3 Report

The paper is written very well, it is clear and concise. I appreciate authors for screening and optimizing different solvents. I have a question for authors i.e. When you use ACN as solvent did authors observe any amide formation (attack of N from ACN molecule to the 1,2-anhydro glucose 3) in addition to thioaryl product.

Author Response

Answers to the reviewer:

We did not observe any attacking of N from ACN molecule to the 1,2-anhydro glucose. However, we did observe a coupling reaction between acetonitrile and BnBr in the presence of sodium hydride in the process of synthesizing tri-OBn glycals.